# Excited-state vibration-polariton transitions and dynamics in nitroprusside

Andrea B. Grafton[1], Adam D. Dunkelberger[2], Blake S. Simpkins [2], Johan F. Triana [3], Federico J. Hernández [4], Felipe Herrera[3] & Jeffrey C. Owrutsky [2 ✉]

Strong cavity coupling to molecular vibrations creates vibration-polaritons capable of modifying chemical reaction kinetics, product branching ratios, and charge transfer equilibria. However, the mechanisms impacting these molecular processes remain elusive. Furthermore, even basic elements determining the spectral properties of polaritons, such as selection rules, transition moments, and lifetimes are poorly understood. Here, we use two-dimensional infrared and filtered pump–probe spectroscopy to report clear spectroscopic signatures and relaxation dynamics of excited vibration-polaritons formed from the cavity-coupled NO band of nitroprusside. We apply an extended multi-level quantum Rabi model that predicts transition frequencies and strengths that agree well with our experiment. Notably, the polariton features decay ~3–4 times slower than the polariton dephasing time, indicating that they support incoherent population, a consequence of their partial matter character.

[1] National Research Council Post-Doctoral Scholar, Washington, DC, USA. [2] Chemistry Division, Naval Research Laboratory, Washington, DC, USA. [3] Department of Physics, Universidad de Santiago de Chile, Santiago, Chile. [4] Department of Chemistry, School of Biological and Chemical Sciences, Queen Mary University of London, London, UK. ✉email: jeff.owrutsky@nrl.navy.mil

I t is widely known that a confined optical mode can couple to a resonant material transition and lead to enhanced photon absorption/emission rates, excited-state population control, and the formation of new hybrid light–matter states (polaritons)[1,2]. Only recently has the field of cavity coupling turned its attention to the modification of chemical reactivity generating exciting results, such as suppressed photoisomerization[3] and photobleaching[4], which result from photonic mode coupling to an electronic transition. Although there is a rich history of coupling to electronic transitions[5–10], coupling to molecular vibrational transitions has only been realized in the last few years[11,12]. Strong coupling between infrared-active vibrational modes and photonic cavities offers the unique possibility of systematic modification of the vibrational energy landscape of a molecular system and specific bond activation/deactivation. This has been demonstrated with reduced bond cleavage rates[13], enhanced enzymatic activity[14], charge transfer equilibria[15], and modified product branching ratios[16]. Key to these processes, we believe, are the hybrid light–matter polariton states formed under strong coupling conditions[17]. Although the energies and dispersive properties of these states can be analytically predicted[6,10], their population dynamics, excitation conditions, and excited-state transition energies and moments are not well understood.

We have previously examined the relaxation dynamics of the triply degenerate CO stretching band of tungsten hexacarbonyl ($W(CO)_6$) coupled to a cavity mode using infrared pump–probe spectroscopy[18,19] and two-dimensional infrared (2D IR) spectroscopy[20]. The hallmark of strong coupling is the formation of two polariton modes, the lower polariton (LP) and upper polariton (UP), each with partial molecular and photonic character. In our initial reports, we found that the transient spectra were dominated by the response of a reservoir of uncoupled excited vibrational states near the LP, which leads to a large absorptive feature there and a characteristic shift of the UP. There was also evidence that the LP spectral region includes polariton-related transitions[18,20–25], but it was difficult to distinguish reservoir and polariton excitations. Furthermore, the extremely strong and narrow vibrational bands of $W(CO)_6$ can lead to efficient population of both the first and second vibrational excited states[26], further congesting the excited-state spectra. In addition, the degeneracy of this CO mode leads to rapid reorientation that may compete with short-lived polariton processes. Despite these challenges, nonlinear infrared spectroscopy of $W(CO)_6$ has continued to yield fascinating results[23,25], including demonstrations of remote communication between distinct optical cavities and strong optical nonlinearities that arise from coherence exchange[22,24] and, very recently, intermolecular vibrational energy exchange[27].

We seek to avoid the ambiguities of peak assignment described above by applying 2D IR to cavity-coupled sodium nitroprusside (SNP). The vibrational spectroscopy, dynamics, photo-chemistry[28–33], and implementation of SNP in optical devices[34–41] have been studied; furthermore, SNP is a prototypical system for studying photoinduced metal-nitrosyl linkage isomerism[33]. Critical to this work, SNP has a similar molecular structure to $W(CO)_6$ and $Fe(CN)_6^{3-}$ but with a nondegenerate NO stretching vibrational mode suitable for strong coupling (Fig. 1 inset)[42]. Moreover, SNP has larger nuclear anharmonicity ($24.8\ cm^{-1}$) than $W(CO)_6$ ($15\ cm^{-1}$) and its absorption strength is somewhat weaker, potentially reducing the spectral congestion near the LP induced by vibrational ladder climbing. Our earlier work on solvent-dependent vibrational relaxation dynamics of SNP[43] revealed that the orientational relaxation time ($T_r = 46\ ps$) is considerably longer than the population relaxation time ($T_1 = 29\ ps$) for the NO band of SNP in methanol (MeOH). This is the inverse of what is seen in $W(CO)_6$, which has an orientational

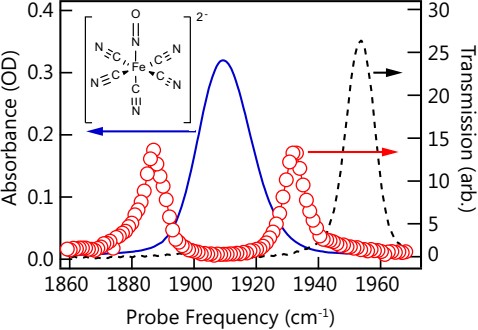

**Fig. 1 Molecular structure and static spectra of SNP.** The blue trace (left axis) is an absorption spectrum of SNP in MeOH in a $CaF_2$ transmission cell. The red trace (right axis) is the static transmission of the ultrafast infrared probe pulse through a 25 µm Fabry–Pérot cavity tuned to resonance with the NO stretching band. Open circles represent the pixel spacing of the detector. The black trace (right axis) is the transmission of the same cavity when the cavity mode is detuned from resonance with the NO stretching band. Inset is schematic of the SNP molecule.

relaxation time (2 ps) nearly two orders of magnitude shorter than its population lifetime (140 ps)[26,44–46]. This distinction has potential implications for cavity-modified anisotropy decay, although we do not address this in the current work. Furthermore, Khalil et al.[47] reported that the homogeneous pure dephasing time of the NO band of SNP is not solvent dependent and is on the order of 2 ps.

In this work, we strongly couple the NO band of SNP to an optical mode of a Fabry–Pérot cavity. Using 2D IR and pump–probe spectroscopies, we apply either spectrally broad coherent excitation, which results in coherent Rabi oscillations, or spectrally filtered excitation to selectively excite either the LP or UP, which suppresses the oscillations and accentuates spectral features associated with polariton excited states. We find polariton excited-state lifetimes are longer than the coherence time revealed by Rabi oscillation decay, providing a signature of the mixed light–matter character of the polariton states. These experimental results are accompanied by a multi-level quantum mechanical model of the ladder of vibrational polariton states within the cavity, which predicts the presence of polariton-related transmissive excited states that agree with experiment.

## Results

**Steady-state infrared transmission.** The static spectra of SNP in and out of the cavity are shown in Fig. 1. The solvent-subtracted absorption spectrum of the uncoupled NO stretching band of SNP in MeOH (solid blue) reveals a band centered at $1909\ cm^{-1}$, a full width half maximum (FWHM) of $20.7\ cm^{-1}$, and integrated band strength of $\sim35{,}000\ M^{-1}\ cm^{-1}$[43]. The transient spectrum of uncoupled SNP (see Supplementary Fig. 1) shows a ground-state bleach at $1909\ cm^{-1}$ and excited-state absorption at $1884\ cm^{-1}$ corresponding to the first vibrational excited state (1–2 transition). It is worth noting that in the uncoupled system, only peak amplitudes vary with changes in populations; the peak positions remain fixed. This is not the case for cavity-coupled systems where spectral shapes can be distorted as has been shown[18,20] and will be discussed further below. For the present work, a 25 µm pathlength cavity was assembled and tuned to the NO stretching band of SNP. The cavity-coupled transmission spectrum in Fig. 1 (red circles) shows two resonant transmission peaks, corresponding to the UP ($1933\ cm^{-1}$) and LP ($1888\ cm^{-1}$) polaritons with linewidths of $\sim8\ cm^{-1}$ and separated by a Rabi splitting ($\Omega$) of $\sim45\ cm^{-1}$. The bare cavity mode linewidth was measured by detuning the cavity from the NO band and found to be $11\ cm^{-1}$

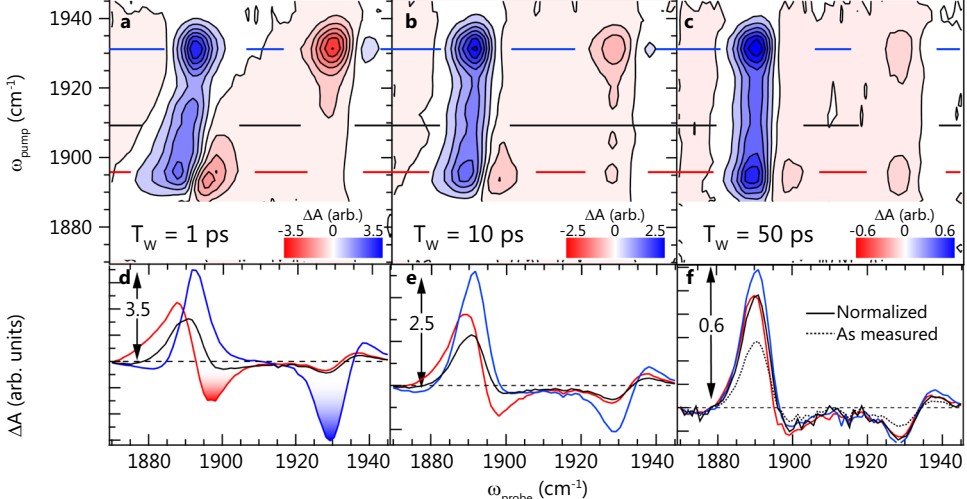

**Fig. 2 2D IR spectroscopy of cavity-coupled SNP.** Two-dimensional infrared spectra of the NO stretching band strongly coupled to an optical mode of a 25 μm-long Fabry–Pérot cavity measured at **a** 1 ps, **b** 10 ps, and **c** 50 ps waiting times. The intensity scale is in units of normalized $-\log(T/T_0)$, with blue positive (decreased transmission) and red negative (increased transmission). Solid colored lines represent the pump frequencies used for the spectral cuts in panels **d**, **e**, and **f**. The blue trace is taken where the pump frequency coincides with the UP, the black trace corresponds to the reservoir, and the red trace corresponds to the LP. In addition, **f** shows the normalized reservoir slice (solid black trace) and the unnormalized reservoir slice (dashed black trace). The Rabi splitting in this data is 35 cm$^{-1}$.

(black dashed curve). The splitting is approximately twice and four times the molecular and cavity linewidths, respectively, ensuring the strong coupling criterion is satisfied.

Molecular dephasing times and cavity photon lifetimes will likely influence polariton dynamics, so it is important to carefully consider the relevant linewidths. In the case of SNP, the NO stretch absorption band exhibits a nearly Gaussian shape, which indicates significant inhomogeneous broadening. Homogenous linewidths are difficult to extract in the presence of inhomogeneous broadening. However, 2D IR spectroscopy does allow the homogenous linewidth to be determined through the frequency–frequency correlation function, which has been applied[47] to the NO stretch of SNP in MeOH yielding a pure dephasing time, $T_2^*$, of 1.7 ps. This value, coupled with the bandwidth product for a Lorentzian pulse (0.32), gives a predicted homogenous linewidth of ~6.3 cm$^{-1}$. Theory predicts that the width of strongly coupled polariton modes, $\Gamma_{\rm CM}$, is the average of the cavity mode width, $\Gamma_{\rm cav}$, and the homogenous vibrational bandwidth, $\Gamma_{\rm H}$[48], as we have previously reported for the cavity-coupled CO band of polymeric PMMA[12]. Thus, the average of our cavity width of 11 cm$^{-1}$ and homogeneous vibrational bandwidth of 6.3 cm$^{-1}$ results in a $\Gamma_{\rm CM} = 8.7$ cm$^{-1}$, which agrees reasonably well with the measured width of 8 cm$^{-1}$, and is substantially narrower than the 20.7 cm$^{-1}$ SNP NO band measured outside the cavity.

**Nonlinear spectroscopy of cavity-coupled SNP.** The 2D IR measures the third-order nonlinear optical response function of the sample providing details on interstate interactions and coherences, as well as their dynamics. In this work, we take special advantage of the ability of 2D IR to elucidate the pump-frequency dependence of transient spectra. Figure 2a–c show 2D IR contour plots for cavity-coupled SNP, with $\Omega = 40$ cm$^{-1}$, at three waiting times, 1, 10, and 50 ps, with pump scatter subtracted off at late waiting times. Spectral cuts at pump frequencies corresponding to the LP, UP, and fundamental 0–1 reservoir molecular transition (i.e., $\omega_{\rm pump} = \omega_{\rm LP}$, $\omega_{\rm UP}$, and $\omega_{01}$) are shown in Fig. 2d–f (each curve is a section taken along the corresponding colored horizontal line in the contour plots). Some

features we observe are easily explained and consistent with our previous results[20]. We begin with a discussion of the cut at $\omega_{\rm pump} = \omega_{01}$ (black, Fig. 2d–f). The section taken at this pump frequency corresponds to direct excitation of population from the ground state to the $v = 1$ reservoir. As the population in $v = 0$ is reduced, the Rabi splitting contracts slightly, resulting in a derivative-like shape where $\omega_{\rm probe}$ is close to $\omega_{\rm UP}$. A mirror image of this derivative feature would appear near the LP as well, were it not for the excited-state absorption from $v = 1$ to $v = 2$, which manifests as a strong positive signal near $\omega_{\rm LP}$. This behavior closely matches what we and others have measured and predicted for W (CO)$_6$[18,21,49]. A simple analytical model qualitatively captures this reservoir-dominated behavior in SNP (see Supplementary Note 1 with Supplementary Fig. 2). We point out that in the language of conventional 2D IR spectroscopy, these features all occur "off" the diagonal, but they should not be considered "cross-peaks" in that they do not report on interactions between distinct modes. The features occur off the diagonal because even small fractional population of $v = 1$ induces spectral responses that span an 80 cm$^{-1}$ probe window.

Key to this work, we observe strong negative features (red peaks Fig. 2a and highlighted negative peaks Fig. 2d) when exciting either polariton. These features are not captured by a model that only includes excited-state reservoir population. Cuts taken through each polariton excitation wavelength result in a strong negative feature somewhat near that respective polariton. Specifically, at early waiting times (Fig. 2d), when $\omega_{\rm pump} = \omega_{\rm UP}$ (blue), a strong negative feature occurs at 1925 cm$^{-1}$, and when $\omega_{\rm pump} = \omega_{\rm LP}$ (red), a corresponding negative feature appears at 1895 cm$^{-1}$. At these early times, the response near $\omega_{\rm probe} = \omega_{\rm LP}$ is more complex when directly exciting the polaritons ($\omega_{\rm pump} = \omega_{\rm UP}$ or $\omega_{\rm LP}$) than when exciting only the reservoir ($\omega_{\rm pump} = \omega_{01}$). This suggests that, while excitation at the reservoir frequency generates only $v = 1$ reservoir population, excitation at either polariton generates both reservoir and polariton excitations, and that the strong negative spectral features are signatures of polariton excitations. Conveniently, the response observed at $\omega_{\rm probe} = \omega_{\rm UP}$ is fairly simple. The nuclear anharmonicity of SNP dictates that higher-lying vibrational absorptions occur at $\omega < \omega_{01}$; thus, we can

rule out their contributions to the negative feature at 1925 cm$^{-1}$ (blue shaded region, Fig. 2d), which might otherwise interfere with identifying it as an excited-state polariton feature. The highlighted negative features and the complex spectrum near $\omega_{LP}$ evolve with waiting time such that late-time spectra for all cases closely match one another. Indeed, at later times (>20 ps), the LP and UP spectral cuts are nearly identical to the reservoir cut (aside from amplitude), suggesting that at later times, only population in the $\nu = 1$ reservoir remains. Therefore, we propose that the strong negative features observed at early times at the LP and UP frequencies are direct signatures of excited polariton states and, next, will support this argument with their time-dependent response and theoretical results predicting their transition energies and transition moments.

For the following results, we chose to obtain pump-probe data at a series of waiting times rather than collecting full 2D IR spectra, which reduced data collection time by two orders of magnitude while still preserving comparable signal-to-noise ratios. We maintain state-selective excitation by using spectrally filtered pump pulses, which has been used before to suppress oscillations in 2D IR measurments[25,50,51]. The spectrally filtered pulses are created with the mid-IR pulse shaper, which allows us to impose an arbitrary frequency mask on the pump pulse. By simply passing only the high-energy half of the pulse (blue, Fig. 3a) or low-energy half of the pulse (red, Fig. 3a), we generated half-pulses referred to as "UP Pump" or "LP Pump." We can also

apply a narrow-band frequency mask to create a "Reservoir-Pump" pulse, albeit with some degradation in time resolution (see Supplementary Fig. 3). Previous reports have shown early-time oscillations[20] and, very recently, Yang et al.[25] confirmed that the oscillations arise from coherence exchange between modes. By using half-pump pulses, as described above, we were able to suppress the early-time oscillations of cavity-coupled SNP and recover the oscillations when exciting both polaritons with a broadband pulse. Figure 3b shows the early-time response of our system probed at 1928 cm$^{-1}$ (the negative response in the UP region) when exciting only the LP (LP Pump conditions, red curve) or maskless pumping of both polaritons (black). When the full pump is employed, the transient response reveals oscillations which have a period of 650 fs and a dephasing time of 700 fs based on the results of fitting to a damped sinusoidal wave, as shown in Fig. 3b. This oscillation period matches well to the 45 cm$^{-1}$ Rabi splitting of the coupled system. When an LP Pump mask is applied, there is no excitation at the UP frequency and the oscillations disappear entirely (red, Fig. 3b). We observe a similar oscillation suppression for other rational choices of pump and probe frequencies (see Supplementary Fig. 4). These results imply that the coherence exchange mechanism and the ability to circumvent it, which occurs in $W(CO)_6$, is also present in SNP.

With an understanding of spectrally filtered pumping and its implications on coherent excitation, we now extract polariton spectra and dynamics. Consider the shape of the $\omega_{pump} = \omega_{01}$, or "Reservoir-Pump," spectrum (black curves, Fig. 2d–f). Under the pumping conditions reported here, the shape of the reservoir excited spectrum changes very little over the course of its decay (detailed in Supplementary Fig. 2). This time invariance is in contrast to reports on $W(CO)_6$ whose transient spectra often exhibit notable peak shifts due to significant excitation magnitude (several percent). In the current work, a comparatively small fraction of SNP is excited from the ground state (small fraction of a percent). As, at later times, all of the 2D IR slices closely resemble one another, as well as the calculated reservoir response, we surmise that, at these later times, all of the excited population resides in the $\nu = 1$ reservoir, and that, by subtracting the reservoir response from the early-time spectral slices, we should reveal polariton-specific features. We must first establish a scaling factor between the reservoir-only response and the polariton-pumped responses, as the efficiency of absorption varies as a function of pump frequency, with less absorption at $\omega_{01}$ than at either $\omega_{UP}$ or $\omega_{LP}$. We determine the necessary scaling factors at a time delay of 25 ps, about an order of magnitude longer than any dephasing (coherence) process. At 25 ps, the maximum UP Pump (LP Pump) signal is a factor of 2.3 (3.5) greater than the Reservoir-Pump signal. We multiply the UP Pump and LP Pump spectra by these scaling factors at every delay time and show the resulting spectra at 1 and 25 ps in Fig. 4a, b (see Supplementary Fig. 5 for the 10 ps spectra). Figure 4b testifies to the reliability of this treatment by showing the excellent agreement between the scaled Reservoir-Pump response and the LP- and UP Pump data at late time, when only reservoir excitation remains in both cases. We emphasize that the scalar multiple is extracted from the peak of the spectrum but the scaling brings about a quantitative agreement at all probe frequencies. At early times, clear differences exist (note negative features indicated by arrows in Fig. 4a), which we propose are due to polariton excitations. Although these features were observable in the uncorrected data (Fig. 2d, e), they are expressed much more clearly after scaling and subtraction of the Reservoir-Pump response.

The reservoir-subtracted spectra ($\Delta t = 1$ ps) shown in Fig. 4c exhibit several distinct features, which we will analyze. In either the UP Pump or LP Pump case, a derivative-like feature appears in the difference spectrum where $\omega_{probe}$ is close to $\omega_{LP}$. We

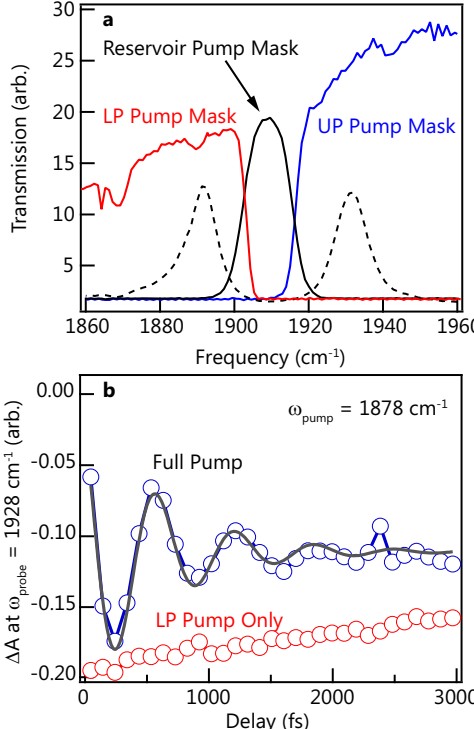

**Fig. 3 Pump filtering masks and pump filtered transient spectra of cavity-coupled SNP. a** Selective-pump masks used to selectively excite either the UP, LP, or reservoir. The blue trace is the spectrum of the pump pulse after filtering with the UP Pump mask; the red trace is the spectrum of the pump pulse after filtering with the LP Pump mask; the black trace is the spectrum of the pump pulse after filtering with a Reservoir-Pump mask. The black dashed line is the static transmission of the probe pulse through the 25 μm Fabry–Pérot cavity. **b** Peak intensity decays of the 2D IR spectrum measured at $\omega_{probe} = 1928$ cm$^{-1}$. The blue trace corresponds to a full spectrum pump (i.e., no filtering), the red trace is measured with the LP Pump, and the gray solid line is the damped sinosoidual fit to the oscillation.

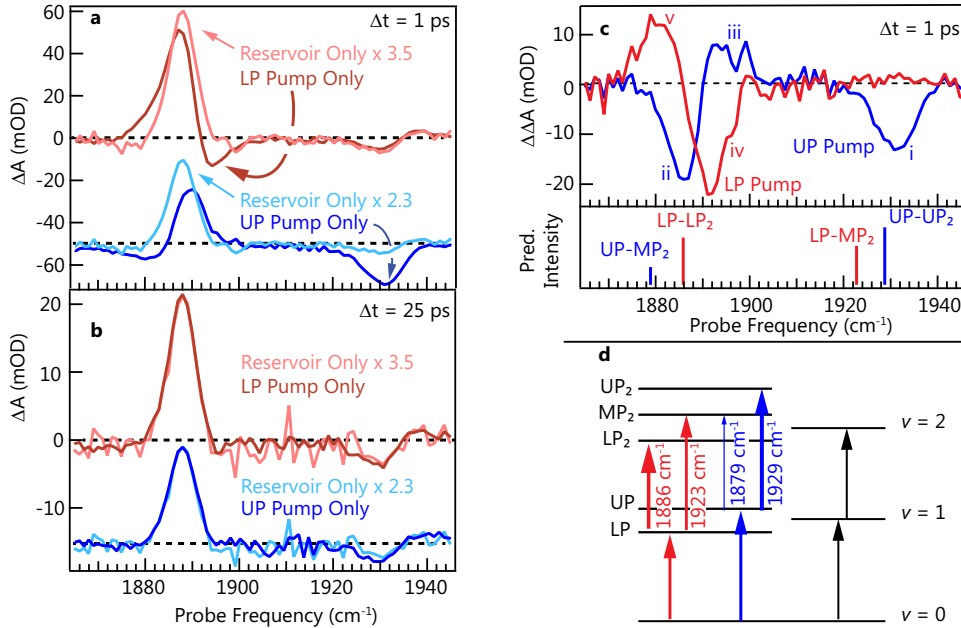

**Fig. 4 Scaled reservoir and subtracted reservoir spectra and theoretical predictions of transitions of cavity-coupled SNP. a** Pump–probe spectra obtained with spectrally filtered pump pulses reported in terms of $-\log(T/T_0)$. The dark red trace is the transient spectrum recorded 1 ps after exciting with the LP Pump mask and the light red trace is the spectrum recorded 1 ps after exciting with the Reservoir-Pump mask then multiplied by 3.5, as described in the text. The blue traces (offset for clarity) are the corresponding spectra for UP Pump mask excitation, for which the scaling factor is 2.3. Rabi splitting is 46.5 cm$^{-1}$. **b** Pump–probe spectra obtained as described in **a**, but with a time delay of 25 ps. The scaled Reservoir-Pump spectra (light red, light blue) overlap the UP Pump (dark blue) and LP Pump (dark red) spectra. **c** Subtracted spectra for UP Pump (blue) and LP Pump (red) excitation. The positive and negative feature frequencies are marked as follows: i = 1932 cm$^{-1}$, ii = 1886 cm$^{-1}$, iii = 1892 cm$^{-1}$, iv = 1892 cm$^{-1}$, and v = 1883 cm$^{-1}$. The stick spectra correspond to the predicted transition frequencies from the LP (red) and UP (blue) to the second excited polariton manifold, with the relative heights proportional to their predicted intensities. **d** Schematic polariton level diagram up to the second excited manifold. Vertical arrows indicate the dominant polariton transitions with the corresponding transition frequencies in cm$^{-1}$. The size of the arrowhead is roughly proportional to predicted transition strengths.

suspect that part or all of this feature corresponds to polariton-involved transitions, but attempts at assignment are hindered by the spectral congestion in this region, as described above. However, the UP Pump spectrum shows a clearly resolved negative feature (i) centered at 1932 cm$^{-1}$. As this frequency is far removed from any higher-lying vibrational transition of the molecule and appears only when the UP is pumped, we conclude that it arises from an excited-state transition from the UP. Reliable assignment of the features requires a rational model for predicting excited-state transition frequencies and amplitudes. We choose to employ a recently published extension of the multi-level quantum Rabi (MLQR) model[52], which takes into account the permanent dipole moments of the bare anharmonic vibrational states[53]. We note that the calculated equilibrium dipole moment of SNP in MeOH exceeds 10 Debye in free space (see Supplementary Note 2); thus, an intracavity model that only considers transition dipole moments between vibrational states is not expected to give a complete picture of the system.

In Fig. 4d, we show the calculated polariton level diagram obtained for the cavity-coupled SNP system, including the main absorptive transitions from the ground state up to the second excited polariton manifold. The latter is composed of three polariton levels that we denote as LP$_2$, MP$_2$, and UP$_2$, in order of increasing energy. The uncoupled $\omega_{01}$ and $\omega_{12}$ vibrational transitions are also shown for energy comparison. We note that reservoir molecular states are fully uncoupled from the polariton states in our Hamiltonian model; thus, they are ignored. The parameters of the Morse potential for the NO vibrational mode, its electric dipole function, the cavity frequency, and the amplitude of the vacuum field fluctuations are constrained to

reproduce the known transition dipole moments of SNP in MeOH determined from integrated band strengths[43,47] (see Supplementary Note 2 with Supplementary Figs. 6 and 7) and the position of the LP and UP linear transmission peaks. The model then predicts the positions and intensities of the transitions from LP and UP to all the polariton states in the second excited manifold.

The model predicts transitions that agree well with the data and are summarized by the bars in the lower panel of Fig. 4c (bar height corresponds to predicted transition moment). In a transmission geometry, the strength of these transitions would be proportional to the square of the matrix element of the field operator $\hat{a}$[54]. Negative features correspond to gain in transmission ($\Delta A = -\log(T/T_0)$), typically assigned to ground-state bleaching and stimulated emission in free space. In this case, however, the new polaritonic transitions also manifest as gain, because the cavity transmits more light at the new polariton frequencies. Considering the transitions predicted to be strongest, the negative feature (iv) in the LP Pump spectrum of Fig. 4c corresponds with the dominant LP to LP$_2$ (observed at 1892 cm$^{-1}$ and predicted to be 1886 cm$^{-1}$). The negative peak (i) in the UP Pump spectrum is assigned to the UP to UP$_2$ transition (observed at 1932 cm$^{-1}$ and predicted to be at 1929 cm$^{-1}$). From either the LP or UP, in addition to the dominant peak, the model also predicts a second satellite transition to MP$_2$. These satellites are predicted to be weaker than the primary LP-LP$_2$ and UP-UP$_2$ transitions for all the conditions tested (see Supplementary Note 2, including predictions for the results shown in Fig. 2). The low-frequency negative feature (ii) in the UP Pump spectrum in Fig. 4c matches the predicted UP-to-MP$_2$ satellite, although its strength relative to

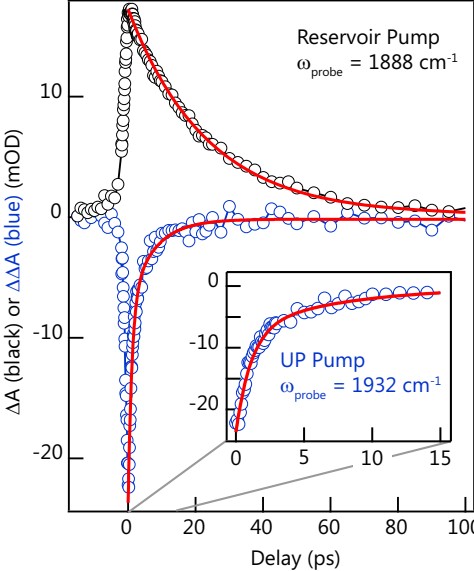

**Fig. 5 Kinetics of the decay of the transient response of cavity-coupled SNP.** The black circles are measurements of the transient response of the cavity-coupled system at 1888 cm⁻¹, the peak of the reservoir excited-state absorption, as a function of time delay after exciting with a Reservoir-Pump filter. The red line is a fit to an exponential decay with a 23 ps lifetime. The blue circles are the subtracted response measured at 1932 cm⁻¹ (i in Fig. 4c) as a function of time delay after exciting with a UP Pump filter. The corresponding red line is a fit to a biexponential decay with time constants 1 and 8 ps.

the higher frequency UP-UP$_2$ feature does not match the model prediction. We see no evidence of the LP-MP$_2$ transition; continued study is required to understand the link between the calculated and observed intensities or lack thereof.

With our proposed assignments in hand, we turn to the kinetics of cavity-coupled SNP. First, we examine the Reservoir-Pump response. Figure 5 shows the temporal evolution of the peak of the Reservoir-Pump response at 1888 cm⁻¹ (associated with reservoir $v = 1$ population, black data). We fit this evolution to an exponential decay and find a time constant of 23 ps consistent with the 24 ps lifetime measured in free space under parallel polarization conditions. We have shown previously that, for W(CO)$_6$ in hexane, fits to kinetic cuts through cavity-coupled transient spectra can give unreliable lifetimes[49], and that a better approach is to fit the entire transient spectrum at each delay to an analytical model that accounts for reservoir population. Accordingly, we fit the series of transient spectra obtained under Reservoir-Pump conditions and find a nearly identical decay for the $v = 1$ population (see Supplementary Fig. 2). The agreement between full spectral fits and simple exponential peak decays further confirms that the spectra are not qualitatively varying (e.g., peak shifts) with time likely due to significantly smaller excitation fractions as compared to W(CO)$_6$ experiments. These findings also confirm that decay rates of bare molecular reservoir states are not altered by cavity coupling.

The UP to UP$_2$ feature shows markedly different kinetics (Fig. 5, blue). We fit the evolution of the peak amplitude of the feature at 1932 cm⁻¹ (i) to a biexponential decay and find time constants 1 and 8 ps (see Supplementary Table 6 for fit parameters). We note that the 1 ps time constant is on the order of the instrument response function of the frequency-narrowed Reservoir-Pump response and we estimate an uncertainty of about 1 ps for the longer decay. However, the 8 ps process has important implications for the behavior of the polariton. This lifetime is much longer

than the ~2 ps dephasing time of the polariton modes (or, the dephasing time of uncoupled molecule or empty cavity optical mode). From this observation, we infer that there exists an incoherent population in the UP level. In other words, the UP does not exist only as a coherent superposition of the vibrational transition and optical mode, but may have some incoherent component stemming from its partial matter character. The other features, both positive (iii and v) and negative (ii and iv), decay with indistinguishable kinetics (with decay times 6.5–8 ps, see Supplementary Fig. 8 and Supplementary Table 6), leading us to speculate that the decay time reflects an inherent property of the polaritons in this system. The proximity of the features in the LP Probe region to the reservoir excited-state absorption introduces considerable spectral ambiguity and precludes a definite assignment of those features. If, for instance, the decay reflected energy transfer to an acceptor mode in the solvent, we might expect different kinetics for UP and LP excitations. The specific mechanism of polariton population decay remains a topic for future study. Furthermore, we apply a reservoir subtraction data analysis to the 2D IR spectra in Fig. 2 (see Supplementary Fig. 9 and Supplementary Note 3) and find similar subtracted spectra and kinetics when the unfiltered pump pulse interacts with the sample.

## Discussion
Our assignment of the UP to UP$_2$ transition in cavity-coupled SNP in MeOH is an important step in joining the theory of vibrational polaritons and ultrafast experimental results. In our first report on the transient spectroscopy of vibrational polaritons, we assigned a transition at the low-energy edge of the transient spectrum of cavity-coupled W(CO)$_6$ to a UP to $v = 2$ transition, but more recent works[23] have called that assignment into question. We do find that in the case of SNP, the spectrum in the $\omega_{probe} \sim$ LP region is congested and, accordingly, we are hesitant to make definitive assignments there. However, the observed feature positions and their evolution are strongly suggestive of polariton-driven behavior and fully consistent with our theoretically predicted polariton excited-state transitions. It is important to note that the critical difference between the theoretical approach we report here and either the analytical model or quantum mechanical treatment developed previously is that we consider higher-lying transitions in the polariton manifold. In previous work, we did not observe strong evidence of the UP to UP$_2$ transition in W(CO)$_6$, but combining pump resolution (either through 2D IR or filtered excitation) with reservoir subtraction made measuring and analyzing such responses much more reliable in this work. In the full-pump pump–probe experiment, negative features in the LP and UP regions, along with the congested response at low frequencies, are all integrated over the $\omega_{pump}$ axis, greatly convoluting the observed spectra; however, 2D IR disperses the pump frequencies along the $\omega_{pump}$ axis providing a more detailed energy landscape.

It is worth discussing the inadequacy of the standard usage of cross-peak and diagonal peak to describe the spectroscopy of strongly coupled vibrations. Although a majority of the features we observe occur away from the diagonal of the 2D IR spectra, they do not necessarily report on energy or coherence transfer between disparate excited states. Considering the 2D IR spectrum at 50 ps, e.g., we find that a slice along $\omega_{pump} = \omega_{01}$ can be captured by the analytically calculated spectrum, which includes only $v = 1$ excitation (see Supplementary Fig. 2). One might then consider the features along this slice as entirely "diagonal" in character because they arise only from direct excitation of the reservoir followed by probing the excited-state response of the reservoir. This is in close analogy to how the excited-state

absorption of an uncoupled vibrational mode is considered part and parcel of the diagonal peak. On the other hand, the responses at $\omega_{pump} = \omega_{LP}$ or $\omega_{UP}$ at late time can also be described by pure $v = 1$ excitation, so one could reasonably identify the entire spectrum as "cross-peak" in character, despite the response spanning both diagonal and off-diagonal regions. We suggest that such language may not be useful here, and that one must, instead, carefully consider the origin of each individual spectral feature in the coupled system. By subtracting the reservoir contribution from the filtered pump-2D IR and pump–probe spectra, we can clearly identify which parts of the spectra correspond to polariton or reservoir population, which then dictates what should be considered a true cross-peak.

To summarize, we have reported transient and 2D IR spectroscopy of cavity-coupled SNP. The multidimensionality of 2D IR and filtered pump–probe results lead to the observation and assignment of a UP to $UP_2$ transition, clear evidence of a transition between excited vibration-polariton levels. The dynamics were measured with filtered pump–probe measurements in which the reservoir-pumped response could be removed to isolate polariton-related signals, which decay on a timescale significantly longer than the polariton dephasing time and shorter than the vibrational population lifetime outside the cavity. This result indicates that vibrational polaritons are not purely coherent phenomena and have incoherent lifetimes that are affected by strong coupling. This work connects vibrational polaritons to exciton polaritons, where the LP can also persist well beyond coherence decay lifetimes of the system[55]. Although the results reported here are important in the sense that distinct coherent and incoherent polariton lifetimes are identified, the specific mechanisms of polariton decay are not clear. Identifying the relaxation processes are, of course, key to developing approaches to controlling these lifetimes. In excitonic systems, examination of long-lived polariton population has revealed important decay channels[2,55,56], including channels by which reservoir excitations relax into the LP. Such a process could make the LP appear to have a longer lifetime due to its being repopulated. However, our data suggest little or no decay from reservoir dark states to a polariton state. This is clear when examining Fig. 5, and Supplementary Figs. 1 and 2, which show reservoir kinetics within the cavity that are identical to that exhibited by the uncoupled molecule. We hypothesize that polariton coherence and population transfer are important for enabling energy transfer processes, such as those reported in ref. 27, and influence the decay of transition state species, which could be a mechanism at work in modified chemical reactivity. Polariton decay is certainly a prominent concern when forming polariton condensates, which has yet to be achieved with vibrational polaritons. Additional studies of polariton relaxation, which aim to contribute to these areas, are currently under way.

## Methods

**Sample preparation.** SNP ($Na_2Fe(CN)_5NO\cdot2H_2O$) and MeOH (>99.9% pure, spectroscopic grade) were purchased from Sigma Aldrich and were used without further purification. Solutions of saturated and nearly saturated SNP in MeOH were prepared, filtered, and subsequently injected into a demountable FTIR Harrick cell containing two dielectric mirrors with ~94% reflectivity (Universal Thin Film Lab Corp.) separated by a 25 µm Teflon spacer. The etalon formed by this empty cavity had a free spectral range ~150 cm$^{-1}$, fringe width of ~10 cm$^{-1}$, and finesse of 15. Filling the cavity with the saturated SNP solutions gives a Rabi splitting, $\Omega \sim 35\text{--}45$ cm$^{-1}$. The Rabi splittings varied due to using slightly different concentrations among our samples. The strong coupling criterion was satisfied for all cavity experiments (i.e., $\Omega$ is greater than the FWHM of the cavity (10 cm$^{-1}$) and SNP molecular band (20 cm$^{-1}$)). Absorption spectra of SNP in MeOH were obtained with a diluted solution using a Thermo Fourier transform infrared (FTIR) spectrometer with a 1 cm$^{-1}$ resolution. The sample is held in a Harrick demountable FTIR cell with $CaF_2$ windows and a 50 µm-thick Teflon spacer under ambient conditions.

**2D IR and transient absorption pump–probe spectroscopy.** The 2D IR experiments are performed with a commercial Ti:Sapphire laser system (Coherent) that outputs 800 nm pulses at a 1 kHz repetition rate with a 35 fs pulse duration and 7 W average power. The mid-IR pulses are obtained by sending 3 W of 800 nm light to pump a commercial optical parametric amplifier (Light Conversion). The signal and idler are spatially and temporally overlapped and are difference frequency mixed in a Type II AgGaS$_2$ crystal to produce ~10 µJ mid-IR pulses that are ~100 fs in duration and centered at 5.2 µm with a FWHM of 115 cm$^{-1}$. A 3° CaF$_2$ wedge splits the mid-IR light into pump and probe pulses. The pump beam is sent through a commercial Germanium acousto-optic modulator (AOM)-based pulse shaper (PhaseTech) generating a pump beam pair. The pulse shaper employs frequency-domain shaping to create two pulses in the pump beam and scans the time delay between them, τ. In addition, the pulse shaper controls the relative phases of the two pump pulses, which allows for phase cycling. Phase cycling isolates the signal from the background, removes scattered light contributions, subtracts transient absorption signals and shifts the signal to the rotating frame to allow a larger fully sampled step size[57–59]. The probe beam propagates to a mechanical delay stage fitted with a retroreflector to control the time delay ($T_w$) between the pump and probe beams. A 90° off-axis parabolic mirror focuses the pump and probe beams into the sample to approximate beam diameters of 150 and 100 µm, respectively. A second parabolic mirror collimates the probe beam after the sample and sends it to the detector. Using a pump–probe geometry, the pump and probe pulses are overlapped spatially and temporally at the sample. The pump beam provides the first two electric-field interactions and the probe beam is the last electric-field interaction. The sample is placed normal to the incident pumping and probe beams (i.e., $k_\parallel \sim 0$) and tuned via lateral translation. The detection system disperses the probe beam onto a 2D-MCT (PhaseTech) collecting an entire spectrum for every laser shot. A ZnSe lens focuses the 2D IR signal into a 250 mm focal length spectrometer. The entire optical table is purged with nitrogen to eliminate water vapor. Transient absorption data are collected on the same setup by setting $\tau = 0$ fs. Our current implementation of 2D IR does not permit access to the reflected probe beam so we are unable to measure the transient absorption, from which the transient absorption of the sample can be computed. However, we have shown previously[18,49] that the transient absorption of the sample gives similar information to the transient transmission for cavity-coupled W(CO)$_6$ and expect this to hold true for this system as well.

**Pump filtering method.** Selective pumping is an established technique[25,50,51], which we implement in the 2D IR apparatus via a pulse shaper. The pulse shaper disperses the pump beam in frequency across the AOM. Using an arbitrary waveform generator, we create custom masks using the PhaseTech QuickControl software to select the frequency range of the pump pulse allowing selective excitation of the LP or UP. It is noteworthy, however, that selective pumping necessarily broadens the pulse in time. Cross-correlations of the pump and probe were measured (see Supplementary Fig. 4) and fit to Gaussian functions to determine the instrument response, which is 0.25 ps for the full pump, 0.4 ps for half pump, and 1.3 ps for the narrow reservoir-only pump. Careful attention was paid to the waiting time, $T_w$. At very early $T_w$, pump and probe pulses are temporally overlapped and generate nonresonant signals, which can distort the 2D band shape. Thus, all reported dynamics are based on measurements taken after $T_w = 0.5$ ps.

**Theoretical methods.** Spectral assignments for the 2D IR polariton spectra are done based on an extension of the MLQR model[52] that takes into account the permanent dipole moments of vibrational states under light–matter coupling[53]. The NO stretching mode of SNP is modelled as a Morse oscillator with reduced mass $\mu = 14.73$ a.m.u., binding energy $D_e = 0.3345$ a.u. and anharmonicity parameter $a = 1.77$ a.u. The potential has 75 vibrational bound states. With these Morse parameters, the predicted $v = 0$ to 1 (hereafter abbreviated 0–1) and 1–2 bare vibrational transition frequencies are 1913 and 1888 cm$^{-1}$, respectively, coinciding with the experimentally measured values within 4 cm$^{-1}$. The electric dipole function along the NO stretch coordinate is modelled as a Rayleigh distribution function with constant bias[52]. The model function is parametrized to closely match the shape of the dipole function obtained using density functional theory calculations with implicit solvent (CAM-B3LYP/6-311++G**/MeOH). The parametrization reproduces well the magnitude of the transition dipole matrix elements for the 0–1, 0–2, and 1–2 vibrational transitions extracted from the IR absorption spectrum of SNP in polar solvents outside the cavity[43] as described in the Supplementary Note 2. After constraining the molecular model this way, the only free parameters of the problem are the cavity detuning from the bare 0–1 vibrational resonance and the magnitude of the vacuum field fluctuations at the cavity frequency.

To model the transient absorption spectra, we set the cavity resonance frequency and vacuum fluctuation strength to best match the measured LP–UP splitting within 2 cm$^{-1}$. After fixing the system Hamiltonian parameters, we compute the polariton energies and states by direct diagonalization including the lowest 30 bound Morse levels and 30 cavity Fock states. The frequencies and strengths of the transitions between polariton eigenstates are obtained using input–output theory in the Schrodinger picture[54]. Transition strengths are proportional to the square of the matrix element of the cavity field operator $\hat{a}$

connecting the initial and final polariton states. Coherent or dissipative couplings between polariton states and dark collective molecular states are neglected. Additional details of the theoretical methods are found in Supplementary Note 2.

## Data availability

The data that support the findings of this study are available on request from the corresponding author J.C.O. upon reasonable request.

## Code availability

The code that supports the findings of this study is available from the corresponding author upon reasonable request.

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

## Acknowledgements

A.B.G., A.D.D., B.S.S., and J.C.O. were supported by the Office of Naval Research through internal funding at the U.S. Naval Research Laboratory. A.B.G. acknowledges the NAS-RAP for administering their fellowships. J.F.T. is supported by ANID through the Postdoctoral Fellowship Grant No. 3200565. J.F.T., F.H., and F.H. were supported by ANID through the Proyecto REDES ETAPA INICIAL, Convocatoria 2017 number REDI 170423, FONDECYT Regular number 1181743, and ANID - Millennium Science Initiative Program ICN17-012.

## Author contributions

All authors contributed equally to manuscript preparation. A.B.G. and A.D.D. carried out the 2D IR and pump–probe measurements and analysis. B.S.S. developed and applied the analytical treatment and advised on interpretation. J.F.T., F.H., and F.H. developed and applied the theoretical treatment and advised on interpretation. J.C.O. oversaw the project and advised on experimental design and interpretation.

## Competing interests

The authors declare no competing interests.
