## [Peer Review File · Nature Communications]

Reviewer #1 (Remarks to the Author):

The authors report on the excited vibro-polariton dynamics of nitroprusside using 2D IR spectroscopy. They find that by the choice of this molecule, they are able to understand better the internal dynamics and vibro-polariton ladder in such strongly coupled system. With the help of a theoretical analysis they are able to assign the transient peaks to upper excited polaritons which is a first. Secondly, they find that the lifetime of the polaritonic state can be much longer than the dephasing time, something that has already been seen for exciton polaritons. The paper is highly technical but well written for a people familiar with light-matter strong coupling. I have number of remarks which the authors need to consider:

- 1) While the data is clean, it assumes that the changes in reflectivity can be ignored, something that is not clear with highly reflective cavities, which in turn can induce large spectral changes when it is taken into account, as first shown by Schwartz et al (Chem. Phys. Chem. 2013).
- 2) It is not clear how the cavity is tuned. Is it at normal incidence ($k=0$) or not? This has critical consequences on the internal dynamics. The system should be at normal incidence to ensure the coupled system doesn't slide down the polaritonic branches.
- 3) The author should discuss at the end the fact that longer population lifetimes (than the dephasing time) has also been observed for exciton-polaritons. Their work and approach could help clarify this issue.
- 4) In the abstract, the authors mention charge transfer equilibria in the abstract, probably referring to the latest paper by the Strasbourg group (Pang et al, Angewandte Chemie 2020). This should be added to the reference list since that work reveals the importance of symmetry on the chemical effects of vibrational strong coupling. In this context it would have been nice if the authors discuss symmetry in more detail beyond just the choice of the molecule on page 2. They could for example give the symmetry type of the coupled mode. Are there any other vibration they could couple of different symmetry?

Reviewer #2 (Remarks to the Author):

The paper reports new results on advanced nonlinear spectroscopy applied to vibration-polaritons. It is important to understand the dynamics and spectroscopy of such polaritons in order to manipulate them, opening new ways in the control of photophysical and photochemical processes. The authors used 2D-IR and pump-probe spectroscopies on cavity-coupled sodium nitroprusside (SNP). They obtained clear evidence of spectroscopic characteristics related to transitions between excited polariton levels and were able to measure polariton decay times, providing new data for further investigation of polariton dynamics. The context is well introduced and the paper is clearly written. The topic is of broad interest, its novelty is obvious, the paper is well suited to Nat. Comm.

However, some points need to be clarified before publication. The comparison with previous results on cavity-coupled $W(CO)_6$ (see ref. 19) should be developed. For example, the derivative-like shape (page 4, line 129) described in the paper is explained in the same way as the case of $W(CO)_6$ in previous studies, whereas in the case of SNP, a very small fraction of the SNP is excited (page 6, line 205), so that the contraction of the Rabi splitting should be reduced and the derivative-like shape difficult to observe. The paper and its conclusions will be more convincing if the authors take into account the following remarks.

- 1) The authors obtained nice 2D IR results (Figure 2) but they mainly discussed the filtered pump-probe results. There seem to be some differences that should be noticed and discussed. The spectra obtained at long waiting times can be compared (Figure 4b and Figure 2f): negative features around 1895 and 1925 cm^{-1} observed in Figure 2f when pumping LP or UP are clearly absent in Figure 4b. Is "the excellent agreement between the scaled Reservoir-Pump response and the LP and UP-Pump data at late time" (shown in Figure 4b) obtained with 2DIR spectra (Figure 2f)? There are cross peaks in the 2D IR spectra that are worth discussing. Is this similar to the results obtained on $W(CO)_6$ (ref.19)? What is the kinetics of these cross peaks revealing the coupling or energy transfers between the polariton modes? It should be interesting to see the results of filtered pump-probe spectra with a 10ps waiting time to compare them with Figure 2e

(for the discussion of cross peaks in particular).

2) The authors mention (page 5) that they have estimated the dephasing time of the polaritons to be 2 ps. What did they measure to obtain this value, or what model did they use to calculate this value?

3) The authors of ref. 19, including some of the authors of the manuscript, used a quantum-mechanical theoretical model to interpret their 2DIR vibration-polariton spectra. What are the results of this kind of model for the system studied in the manuscript? What are the main differences that cannot be reproduced by the model described in ref. 19? Figure 5 in reference 19 shows the differences between the classical and the quantum model, which should also be interesting to discuss with the SNP results.

4) Positive parts in Figure 4c are not analyzed, why?

5) Figure 5 showing the kinetics of polaritons is particularly interesting. The authors should give the complete set of parameters of their fit with two exponential decays, to show the relative weights of the short and the long components (same remark for Supplemental Figure 8). The authors should also compare with the lifetime of the empty cavity mode (less than 2 ps with their experimental parameters). The long component could come from a population of polaritons via energy transfers (suggested by the 2D IR spectra) and the influence of the pump (using different masks) should be studied to inform on a possible dependence of this long component on the pump filter (or to discard it). It would be interesting to obtain similar decays from the full 2D IR spectra to highlight the role of energy transfers (involving the molecular reservoir with its very long lifetime) or coherent pumping of LP and UP in these kinetics. At the very least, get a few points from 2D IR spectra vs waiting time to verify that they are consistent with the fit obtained with filtered pumps. The molecular transition is very broad and the half-pump masks cannot avoid the excitation of uncoupled molecules. Additional data should be exploited before speculating that "the decay time reflects an inherent property of the polaritons in this system".

Minor points:

- Methods: As the manuscript describes experiments involving two Rabi splitting values, details can be added on how to obtain either value (is there a device to tune the Rabi splitting on demand?).
- Page 2, line 64: 2 ps is nearly two orders of magnitude shorter than 140 ps.
- Page 3, line 111: the bandwidth product for a Lorentzian pulse is not 0.15, but 0.32; Is there a reference with this value of 0.15? Taking 0.32 gives a homogeneous vibrational bandwidth of 6.2 cm⁻¹, which gives an average of 8.6 cm⁻¹ instead of 7 cm⁻¹; 8.6 cm⁻¹ agrees reasonably with the measured width of 8 cm⁻¹.
- Page 6 Figure 3a: add the description of the dashed black line in the caption.
- Page 8, first line of the caption in Figure 4: "and theoretical predictions"
- Page 8, line 265: The model predicts transitions that agree well with ... , not very well in fact (also written in the abstract)
- Page 10, line 333: "The multidimensionality of 2D IR results lead to the observation and assignment of a UP to UP2 transition, clear evidence of a transition between excited vibration-polariton levels" is not entirely true, because the analysis come from the analysis of filtered pump-probe experiments.
- Page 11, line 346: missing character between NO and 2H2O
- Page 11, line 367: Additionally
- Supplemental Note 1: the authors could add a reference to justify the values in their model for the fundamental and excited states - center frequencies and linewidths -(1906 cm⁻¹, 1883 cm⁻¹, 5.3 cm⁻¹, 10.6 cm⁻¹); in particular, the linewidth does not correspond to that of the main text (2.9cm⁻¹), but do correspond to a dephasing time of 2ps.
- Supplemental Figure 2: the model predicts a negative part around 1906cm⁻¹, not observed experimentally (experimental points are missing in this range?), what is the explanation?
- Supplemental, line 94: close to 1
- Supplemental Note 2: add references on the computational method for DFT calculations - line 111, PCM is not defined for example; equation S9 uses 5 parameters that seem to be determined from less than 5 data, what did the authors do?
- Supplemental Table 5: the experimental UP2->MP2 can be added.

Reviewer #3 (Remarks to the Author):

This is an excellent paper that explores excited state polariton dynamics in the cavity-coupled NO band of nitroprusside. The authors use a combination of 2DIR and selective pump-probe spectroscopies and support their experimental observations with theoretical work. Interestingly, they find that the polariton features decay on much slower timescales than the polariton dephasing time, which they interpret as indicating that vibrational polaritons are not purely coherent phenomena and support incoherent populations. This has important implications for our fundamental understanding of polaritons and their potential applications. Several relatively minor points should be addressed prior to publication.

1) The authors should show a fit to the Rabi oscillations in Figure 3b, indicating the frequency and dephasing times extracted from their fit.

2) In some of the figures it is difficult to discern the frequencies that are being discussed. For key frequencies such as 1931 cm⁻¹ it would be helpful to indicate, with a light dashed line or arrow, the features being discussed. In Supplemental Figure 3 vertical lines at the peak positions would help the reader see that there are no observable frequency shifts.

3) A key claim in this paper is there exists an incoherent population in the UP level. This claim is supported by the data shown in Figure 5, where the UP to UP2 kinetics are fit to a biexponential with rates of 1ps and 8ps. The authors argue because the 8ps relaxation is considerably longer than the 2ps they report for the polariton dephasing time, there must be an incoherent population in the UP level. In Supplemental Figure 8 they show similar rates for the LP. Why is the emphasis placed on the UP in the main text? Figure 5 should give a more extensive comparison of the various population kinetics.

4) The authors state "The specific mechanism of polariton population decay remains a topic for future study", and "Our new observations may have implications for designing photophysical experiments that rely on excited state population in vibration-polariton modes." While future experiments are surely needed, the authors should share some of their thoughts on these important issues. They should comment on the possible mechanisms of polariton population decay and on the implications of their findings on future applications employing polaritons for controlling excited state populations and modifying chemical reactivity.

Responses to reviews for **Excited-State Vibration-Polariton Transitions and Dynamics in Nitroprusside**

Key: Reviewer, **Response**, *Change in text*

Reviewer 1

1) While the data is clean, it assumes that the changes in reflectivity can be ignored, something that is not clear with highly reflective cavities, which in turn can induce large spectral changes when it is taken into account, as first shown by Schwartz et al (Chem. Phys. Chem. 2013).

Rev. 1; Point 1. The reviewer's comment is well taken, but we have previously shown [see SI's of Nat. Comm. 7, 13504 and JPCA 122, 965] that time-resolved infrared experiments are, apparently, less susceptible to the phenomena that caused the transmission measurements to differ from the absorption measurements by Schwartz, *et al.* More practically, 2D IR is a more demanding technique than pump-probe spectroscopy, and our current setup does not allow for data collection in reflection mode. Indeed, 2D IR reflection is comparatively rare. To address this concern, we have added text to the Methods section (pg 12) that reads, *"Our current implementation of 2D IR does not permit access to the reflected probe beam so we are unable to measure the transient reflectance, from which the transient absorption of the sample can be computed. However, we have shown previously [Ref. 18, 49] that the transient absorption of the sample gives similar information to the transient transmission for cavity-coupled $W(CO)_6$ and expect this to hold true for this system, as well."*

2) It is not clear how the cavity is tuned. Is it at normal incidence ($k=0$) or not? This has critical consequences on the internal dynamics. The system should be at normal incidence to ensure the coupled system doesn't slide down the polaritonic branches.

Rev. 1; Point 2. The reviewer is correct that the angle at which the measurement is made can have important implications for polariton scattering and relaxation and, in fact, is demonstrated by the tuning dependence of cavity-modified chemistry and is critical in the formation of exciton-polariton condensates. In our experiment, we monitor the system near $k=0$ (incident angle ~ 0 degrees).

In acknowledgment of the importance of the reviewer's comment, we have added the following text to the Methods section of the revised manuscript: *"The sample is placed normal to the incident pumping and probe beams (i.e., $k_{||} \sim 0$) and tuned via lateral translation."*

3) The author should discuss at the end the fact that longer population lifetimes (than the dephasing time) has also been observed for exciton-polaritons. Their work and approach could help clarify this issue.

Rev. 1; Point 3. Exciton-polariton systems, which have been studied more extensively than vibration-polaritons, can exhibit lifetimes longer than native dephasing times. We neglected to highlight this fact in the original manuscript, but appreciate the reviewer's recommendation and now include the following discussion with new references (55 and 56) on p. 11: *"In excitonic systems, examination of long-lived polariton population has revealed important decay channels [Ref. 2,55,56], including channels by which reservoir excitations relax into the lower polariton. Such a process could make the lower polariton appear to have a longer lifetime due to its being repopulated. However, our data suggest little or no decay from reservoir dark states to a polariton state. This is clear when examining Figs. 5, S1, and S3, which show reservoir kinetics within the cavity that are identical to that exhibited by the uncoupled molecule."*

4) In the abstract, the authors mention charge transfer equilibria in the abstract, probably referring to the latest paper by the Strasbourg group (Pang et al, Angewandte Chemie 2020). This should be added to the reference list since that work reveals the importance of symmetry on the chemical effects of vibrational strong coupling. In this context it would have been nice if the authors discuss symmetry in more detail beyond just the choice of the molecule on page 2. They could for example give the symmetry type of the coupled mode. Are there any other vibration they could couple of different symmetry?

Rev. 1; Point 4. We appreciate the suggestion to mention the charge transfer equilibria study and have added the phrase, “charge transfer equilibria,” after “enhanced enzymatic activity” on pg 1 line 33 and cite Pang et al (ref 15). The choice of the nondegenerate NO band is one of the salient aspects of our study which have pointed out in the paper. This choice simplifies data interpretation by avoiding the fast anisotropy decay which complicates analysis of our (and now others) previous studies of the degenerate band of $W(CO)_6$. The NO band that we investigate here has A_1 symmetry. The other relevant IR active modes are the two CN bands near 2150 cm^{-1} that are overlapping with E and A_1 symmetries as discussed in Sando et al. (ref. 43 and references therein). It would certainly be interesting so see if the symmetry has an influence on the modified dynamics.

Reviewer 2

The comparison with previous results on cavity-coupled $W(CO)_6$ (see ref. 19) should be developed. For example, the derivative-like shape (page 4, line 129) described in the paper is explained in the same way as the case of $W(CO)_6$ in previous studies, whereas in the case of SNP, a very small fraction of the SNP is excited (page 6, line 205), so that the contraction of the Rabi splitting should be reduced and the derivative-like shape difficult to observe. The paper and its conclusions will be more convincing if the authors take into account the following remarks.

In our interpretation, the derivative-like lineshape does derive from depopulation of the ground state and even a very small excitation fraction can still generate this feature. Our calculations in Supp. Note 1 and Supp. Figure 2 predict an observable derivative-like lineshape even for small population changes and describe the data quite well.

1) The authors obtained nice 2D IR results (Figure 2) but they mainly discussed the filtered pump-probe results. There seem to be some differences that should be noticed and discussed. The spectra obtained at long waiting times can be compared (Figure 4b and Figure 2f): negative features around 1895 and 1925 cm^{-1} observed in Figure 2f when pumping LP or UP are clearly absent in Figure 4b.

Rev. 2; Point 1. Upon surveying our 2D IR data, we find our late waiting times data are highly sensitive to pump beam scatter, which caused negative features to appear in the 2D IR spectra. We have now corrected for this by subtracting the measured pump scatter. The corrected 2D IR spectra, shown in **the revised Fig. 2**, are now in much better agreement with the pump probe results presented in Fig. 4, and illustrate our main conclusions even more convincingly. The effects of this artifact were most pronounced where the pump and probe frequencies are equal, an unfortunate coincidence that gave rise to the differences in the previous figure. A phrase was added to p. 4 indicating that pump scatter was subtracted in Fig.2. It reads, “...with pump scatter subtracted off at late waiting times.”

Is “the excellent agreement between the scaled Reservoir-Pump response and the LP and UP-Pump data at late time” (shown in Figure 4b) obtained with 2DIR spectra (Figure 2f)? There are cross peaks in the 2D IR spectra that are worth discussing. Is this similar to the results obtained on $W(CO)_6$ (ref.19)?

In the original manuscript, our polariton excited state lifetime determination based on isolating it with our reservoir subtraction procedure was only applied to pump-probe data (Fig. 4) and not to the 2D IR data (Fig. 2). Of course, if our conclusions are correct, the same (or similar) subtraction procedure could be applied to the 2D IR data yielding similar results. Applying this data processing procedure to the 2D IR spectra is more complex since we must account for the spectral resolution along the pump axis. However, based on the reviewer's comment, we have carried out this analysis and find that the subtracted 2D IR spectra give very similar spectral and dynamical information to that obtained from the subtracted pump-probe data (fast components are on the order of 1 ps and slow components that average 7.6 ± 1.0 ps). We note that this approach relies on the key assumptions that we can use the signal magnitudes at 25 ps to extract the relative excitation efficiency as a function of pump frequency AND that this relative efficiency does not change over the course of the evolution of the system. Because of these assumptions, we have decided to present the 2D IR subtraction data as Supplementary Material (Fig. S9 and Note 3) rather than including it in the main text. We have also added the following text to the main manuscript on pg 10: *"Furthermore, we apply a reservoir subtraction data analysis to the 2D IR spectra in Fig. 2 (see **Supp. Fig. 9** and **Supp. Note 3**) and find similar subtracted spectra and kinetics when the unfiltered pump pulse interacts with the sample."*

Regarding discussion of the diagonal and off-diagonal peak features, there are off-diagonal peaks present in the 2D IR spectra and, as we stated in the original manuscript p. 4 line 135, these peaks *should not be considered cross-peaks* as conventionally discussed in the 2D IR literature. We argue that the terminology of cross-peak or diagonal-peak is fraught with peril in cavity-coupled systems. Consider the 2D IR spectrum at 50 ps in Fig. 2: the probe spectrum can be described solely by reservoir excitation regardless of the excitation wavelength. In one framing, then, all of the features at $\omega_{\text{pump}} = \omega_{01}$ are what one would conventionally call "diagonal" peaks, because the reservoir is excited and the response is that of the reservoir, while all of the features at $\omega_{\text{pump}} = \omega_{\text{LP}}$ or ω_{UP} would be characterized "cross" peaks, since the LP or UP is excited and the response is that of the reservoir. To address this ambiguity in the language, we have added the following to the discussion,

*"It is worth discussing the inadequacy of the standard usage of cross-peak and diagonal peak to describe the spectroscopy of strongly-coupled vibrations. While a majority of the features we observe occur away from the diagonal of the 2D IR spectra, they do not necessarily report on energy or coherence transfer between disparate excited states. Considering the 2D IR spectrum at 50 ps, for example, we find that a slice along $\omega_{\text{pump}} = \omega_{01}$ can be captured by the analytically calculated spectrum, which includes only $\nu = 1$ excitation (see **Supp. Fig. 2**). One might then consider the features along this slice as entirely "diagonal" in character because they arise only from direct excitation of the reservoir followed by probing the excited-state response of the reservoir. This is in close analogy to how the excited-state absorption of an uncoupled vibrational mode is considered part and parcel of the diagonal peak. On the other hand, the responses at $\omega_{\text{pump}} = \omega_{\text{LP}}$ or ω_{UP} at late time can also be described by pure $\nu = 1$ excitation, so one could reasonably identify the entire spectrum as "cross peak" in character. We suggest that such language may not be useful here, and that one must, instead, carefully consider the origin of each individual spectral feature in the coupled system. By subtracting the reservoir contribution from the filtered pump- 2D IR and pump-probe spectra, we can clearly identify which parts of the spectra correspond to polariton or reservoir population, which then dictates what should be considered a true cross peak."*

Lastly, considering the current work in context with the results obtained on $\text{W}(\text{CO})_6$ in Ref. 19 (now Ref. 20), we note that Ref. 19 only discussed data collected at late waiting times (i.e., after all polariton excitations had decayed into reservoir states) and shows no filtered pump results. The 2D IR

spectra of the two molecules studied do show some similar spectral features, such as the molecular dark state reservoir absorption and polariton contraction. However, since no early time data were reported in Ref. 19, there was no evidence of polariton excited-state features as we are currently reporting for the SNP system at early times.

It should be interesting to see the results of filtered pump-probe spectra with a 10ps waiting time to compare them with Figure 2e (for the discussion of cross peaks in particular).

Per the reviewer's request, we have added the 10 ps waiting time pump-probe data to the SI as *Supp. Fig. 5*.

2) The authors mention (page 5) that they have estimated the dephasing time of the polaritons to be 2 ps. What did they measure to obtain this value, or what model did they use to calculate this value?

Rev. 2; Point 2. We originally estimated that it was less than 2 ps, intending it as an upper limit based on the polariton linewidth. The time is expected to be closer to 1 ps based on a Lorentzian time bandwidth product (0.32, now noted on p. 3) and measured linewidths. We now fit the oscillations to a damped sine wave to extract the oscillation period (as recommended by Rev. 3) and dephasing time. We now report these fitting results on p. 5 as, *"When the full pump is employed, the transient response reveals oscillations which have a period of 650 fs and a dephasing time of 700 fs based on the results of fitting to a damped sinusoidal wave, as shown in Fig. 3b."*

3) The authors of ref. 19, including some of the authors of the manuscript, used a quantum-mechanical theoretical model to interpret their 2DIR vibration-polariton spectra. What are the results of this kind of model for the system studied in the manuscript?

Rev. 2; Point 3. The reviewer highlights one of the most important aspects of the current submitted manuscript. The model we present yields energies and transition amplitudes associated with *polariton excited states*. This is not the case for the model presented in the cited reference 19 (now ref 20), which only accounts for excitation residing in "reservoir" modes. The Collet-Gardiner input-output approach with the Hamiltonian in Eq. (S7) of ref. 19 (PNAS 115, 4845, 2018) is unable to predict transitions from the first excited polariton manifold to the second excited polariton manifold, which we clearly observe to occur in cavity-coupled SNP. The model in ref. 19 is tailored to describe the long-delay response of the cavity system (>25 ps), so a key assumption is made in this regard: "the number of populated cavity photons is zero, but a transient [molecular] excited-state population exists". In other words, the possibility of polariton-to-polariton transitions induced by the probe field is neglected by construction of that model. Despite this limitation, the approach in ref. 19 can potentially be used to describe the long-delay response of our SNP cavities in future work. We also note that the analytical expression we use here is arguably better than the QM model at reproducing the experimental data in ref. 19 and provides an excellent fit to the data in this work.

What are the main differences that cannot be reproduced by the model described in ref. 19? Figure 5 in reference 19 shows the differences between the classical and the quantum model, which should also be interesting to discuss with the SNP results.

We use a state-of-the-art quantum electrodynamics model (discussed in detail in refs. 52, 53) to understand the short-time response of anharmonic vibrational polaritons, taking into account the entire Morse spectrum of SNP (improving the quartic perturbation in Eq. S7 of ref. 19), a realistic electric dipole function based on ground state DFT calculations (improving the cubic expansion in Eq. S9 of ref. 19), and counter-rotating light-matter coupling. The latter has been shown to be important for smaller Rabi frequencies than expected when molecular vibrations are highly polar (ref. 53, now in JCP 152, 234111, 2020), as is the case for SNP. The effective interaction Hamiltonian in Eq. S11 of ref.

19, needs to be extended as in refs. 52 and 53 in order to describe the lowest two excited polariton manifolds that are relevant to understand our short-delay response. More specifically, the matrix Eq. S11 in ref. 19 is obtained by truncating the Hamiltonian in Eq. 11 of ref. 52 to the three-dimensional subspace ($v=0, n=1$), ($v=1, n=0$) and ($v=2, n=0$), and replacing the coupling strength g by $\sqrt{N(1-2f)}g$. In our model, we also take into account coupling of the cavity field with the 0-2 overtone, which is not negligible in SNP, and properly include Fock states with $n=2$ photons. The full level coupling structure for the lowest excitation manifold is illustrated in Fig. 4b of ref. 52. To clarify this point, we have added a sentence to the discussion on pg 10 that reads, *“It is important to note that the critical difference between the theoretical approach we report here and either the analytical model or quantum mechanical treatment developed previously is that we consider higher-lying transitions in the polariton manifold.”*

4) Positive parts in Figure 4c are not analyzed, why?

Rev. 2; Point 4. The positive features in Figure 4c overlap with features associated with reservoir population making their analysis somewhat ambiguous. Although their origin is, as yet, unclear, we have added kinetic traces for the positive features to the Supplemental Information and have modified the text on pg 10 to now read: *“The other features, both positive (iii and v) and negative (ii and iv), decay with indistinguishable kinetics (with decay times 6.5 to 8 ps, see Supp. Fig. 8 and Table 6), leading us to speculate that the decay time reflects an inherent property of the polaritons in this system. The proximity of the features in the LP Probe region to the reservoir excited-state absorption introduces considerable spectral ambiguity and precludes a definite assignment of those features. In previous work, we did not observe strong evidence of the UP to UP2 transition in W(CO)₆, but combining pump resolution (either through 2D IR or filtered excitation) with reservoir subtraction made measuring and analyzing such responses much more reliable in this work”*

5) Figure 5 showing the kinetics of polaritons is particularly interesting. The authors should give the complete set of parameters of their fit with two exponential decays, to show the relative weights of the short and the long components (same remark for Supplemental Figure 8).

Rev. 2; Point 5. For completeness, a table of the fit parameters has been added to the SI as well as a brief reference to these parameters on pg. 10, reading *“see Supp. Table 6 for fit parameters”*.

The authors should also compare with the lifetime of the empty cavity mode (less than 2 ps with their experimental parameters). The long component could come from a population of polaritons via energy transfers (suggested by the 2D IR spectra) and the influence of the pump (using different masks) should be studied to inform on a possible dependence of this long component on the pump filter (or to discard it). It would be interesting to obtain **similar decays from the full 2D IR spectra** to highlight the role of energy transfers (involving the molecular reservoir with its very long lifetime) or coherent pumping of LP and UP in these kinetics.

The reviewer’s point is well taken – as described in our response to this reviewer’s Point 1, we have undertaken reservoir subtraction for 2D IR spectra and we find that polariton dynamics extracted from 2D IR data are consistent with those found from pump-probe response. We have added the following text to the manuscript on pg 10 : *“Furthermore, we apply a reservoir subtraction data analysis to the 2D IR spectra in Fig. 2 (see Supp. Fig. 9 and Note 3) and find similar kinetics and subtracted spectra when the unfiltered pump pulse interacts with the sample.”*

The new Supplemental Note 3 reads,

“Subtracting the reservoir-only response from 2D IR spectra is less straightforward than for pump-probe spectra because of the increased dimensionality. One must account for the pump frequency

dependence of the 2D IR response. Our approach is to use the 2D IR spectrum obtained at $T_w = 25$ ps as representative of “late-time” response, just as we considered a 25 ps delay for the pump-probe subtraction, in order to isolate the reservoir-only response. We extract a slice of the 2D IR spectrum (Supp. Fig. 9a) along $\omega_{\text{pump}} = 1909$ cm^{-1} , the peak of the free-space absorption spectrum, to obtain the cyan trace in Supp. Fig. 9b. This trace represents the probe spectrum after exciting only the reservoir. We then extract a slice of the 2D IR spectrum along $\omega_{\text{probe}} = 1892$ cm^{-1} , where the 2D IR signal is maximized, to obtain the pink trace in Supp. Fig. 9b. This trace represents the relative efficiency with which a given pump frequency generates the reservoir-only response. Multiplying these two traces results in a matrix plotted as contours in Supp. Fig. 9b which corresponds to the reservoir-only 2D IR spectrum. For each T_w , we scale this matrix by the ratio of the intensity at $\omega_{\text{pump}} = 1909$ cm^{-1} , $\omega_{\text{probe}} = 1892$ cm^{-1} . The scaling factors we obtain decay exponentially with a 24 ps lifetime, just as we would expect from the reservoir decays in the main text. We observe no indication of modified reservoir kinetics that might indicate energy transfer from the reservoir to either polariton mode. Finally, we subtract each scaled reservoir 2D IR spectrum from the 2D IR spectrum at each T_w . The subtracted spectrum at $T_w = 1$ ps (Supp. Fig. 9c) exhibits derivative-like features in the LP Probe region for both UP Pump and LP Pump. The UP probe region shows a strong negative feature for UP Pump, but no response for LP Pump. These features are in reasonable qualitative agreement with the subtracted pump-probe spectra in the main text.

We hesitate to draw strong conclusions from the subtracted 2D IR spectra because of the many assumptions that underlie the subtraction of the reservoir contribution, but identify two salient points. First, the subtracted features are not strongly stretched along the diagonal axis. We infer that, within a given pump region, the probed response is insensitive to the specific pump frequency. Put differently, there appears to be little frequency-frequency correlation in the subtracted responses. Second, the intensity of the negative features decays with similar kinetics to those obtained with pump-probe spectroscopy (Supp. Fig. 9d). The relative paucity of datasets and time delays from the 2D IR preclude a detailed statistical analysis, but each trace yields a fast component on the order of 1 ps and a slow component that averages 7.6 ± 1.0 ps across the three features in this dataset. This similarity suggests that the presence of pump photons at both LP and UP frequencies does not dramatically change the decay pathways available to the excited system.

The molecular transition is very broad and the half-pump masks cannot avoid the excitation of uncoupled molecules. Additional data should be exploited before speculating that “the decay time reflects an inherent property of the polaritons in this system”.

The reservoir-subtracted 2D IR spectra, described above, also allow us to address this concern. We find that the polariton features that remain after subtraction show very little correlation between pump frequency and probe frequency, even when the pump frequency is far from the center of the uncoupled reservoir absorption. This indicates that unintended direct excitation of uncoupled reservoir molecules is negligible. We address this concern in the added text titled Supplemental Note 3, written in the response above.

Minor points:

Methods: As the manuscript describes experiments involving two Rabi splitting values, details can be added on how to obtain either value (is there a device to tune the Rabi splitting on demand?).

The different Rabi splittings are a result of using slightly different SNP concentrations. Tuning coupling strength “on the fly”, has been achieved but is not common. Potential approaches are electrochemical cycling or fluidic flow of solution with varying concentrations. To clarify, we have added the following text to the manuscript on pg. 11: *“The Rabi splittings varied due to using slightly different concentrations among our samples”*

Page 2, line 64: 2 ps is nearly two orders of magnitude shorter than 140 ps.

Changed as recommended.

Page 3, line 111: the bandwidth product for a Lorentzian pulse is not 0.15, but 0.32; Is there a reference with this value of 0.15? Taking 0.32 gives a homogeneous vibrational bandwidth of 6.2 cm^{-1} , which gives an average of 8.6 cm^{-1} instead of 7 cm^{-1} ; 8.6 cm^{-1} agrees reasonably with the measured width of 8 cm^{-1} .

We thank the reviewer very much for correcting us on this point. We used the value from Diels and Rudolph (2006), which relates the FWHM duration of a pulse to its linewidth. We used both the reviewer’s suggested TBWP and the Fourier transform of a homogenous response function as described straightforwardly by Cundiff and coworkers (Optics Express 2010) to obtain 6.2 cm^{-1} , just as the reviewer suggested. This value, indeed, results in a predicted polariton width of 8.6 cm^{-1} , agreeing quite well with our measured value of 8.0 cm^{-1} . We have recalculated the fits in the new Supplementary Fig. 2, considering these new widths, and find the extracted lifetimes are essentially unchanged. We have corrected the manuscript text on pg. 3 and in the SI to reflect the corrected linewidths. We also removed the original Supp. Fig. 2 since the information was available in the following Figure.

Page 6 Figure 3a: add the description of the dashed black line in the caption.

Changed as recommended.

Page 8, first line of the caption in Figure 4: “and theoretical predictions”

Changed as recommended.

Page 8, line 265: The model predicts transitions that agree well with ... , not very well in fact (also written in the abstract)

We predict the position and relative intensities of polariton-to-polariton spectral lines that match those observed at early times within the linewidths. The predictions are only based on a calibration procedure that forces the model to predict an accurate IR absorption spectrum, including 1-2 transitions and 0-2 overtones, in addition to match the Rabi splitting in linear transmission. This level of predictive power is not available in the vibrational polariton literature. We soften the language in the manuscript to read *“agrees well”* on pg 1 and 8..

Page 10, line 333: “The multidimensionality of 2D IR results lead to the observation and assignment of a UP to UP2 transition, clear evidence of a transition between excited vibration-polariton levels” is not entirely true, because the analysis come from the analysis of filtered pump-probe experiments.

We have clarified the text to read *“The multidimensionality of 2D IR and the filtered pump-probe results lead to...”*. Additionally we have added the dynamics analysis of the 2D IR spectra to the SI (more fully described in response to the first point of reviewer 2).

Page 11, line 346: missing character between NO and 2H2O

Changed as recommended.

Page 11, line 367: Additionally
Changed as recommended.

10 - Supplemental Note 1: the authors could add a reference to justify the values in their model for the fundamental and excited states - center frequencies and linewidths -(1906 cm⁻¹, 1883 cm⁻¹, 5.3 cm⁻¹, 10.6 cm⁻¹); in particular, the linewidth does not correspond to that of the main text (2.9cm⁻¹), but do correspond to a dephasing time of 2ps.

We have revisited our assignments of these linewidths, in part due to the reviewer's earlier comment regarding the correct time bandwidth product, and now use the following parameters for the ground state transition ($\nu_{01}=1906\text{ cm}^{-1}$, $\Gamma_{01}=8.6\text{ cm}^{-1}$), and first excited state transition ($\nu_{12}=1883\text{ cm}^{-1}$, $\Gamma_{12}=21\text{ cm}^{-1}$). The center frequencies are taken directly from fits to our own data (such as that in Supp. Fig. 1). The linewidth of the ground state transition is the homogenous width taken from the Khalil group's work on SNP [Ref 47] and that of the first excited state transition represents the heterogeneous width from the same reference. The use of unbroadened versus broadened lineshapes stems directly from the fact that the ground state transition is strongly cavity coupled (polariton linewidth is insensitive to heterogeneous broadening of the matter excitation) while the paltry population generated in the first excited state transition results only in weak cavity coupling (weakly coupled transitions reside in the cavity-enhanced absorption regime where the resulting spectra reflect the broadened material line). These corrected values and their origins are presented in Supp. Note 1 and incorporated in the fits in Supp.Fig. 2.

We included refs (9 and 10), modified the value quoted and used for $\Gamma_{01} = 8.6$ and added the sentence, *"Specifically, these values correspond to the homogeneous linewidth of the strongly coupled ν_{01} transition and the inhomogeneously broadened width of the excited ν_{12} state since it's in the weak coupling cavity-enhanced absorption regime."*

Supplemental Figure 2: the model predicts a negative part around 1906cm⁻¹, not observed experimentally (experimental points are missing in this range?), what is the explanation?

For frequencies very close to the uncoupled reservoir absorption line, the transmitted probe intensity is below the detection limit of our instrument both with and without the pump pulse present, so $-\log((T-T_0)/T_0)$ diverges and effectively reports on the noise of the baseline of the detector. The calculation, of course, has no detection limit and so suggests that the transmission should slightly increase at the peak of the uncoupled absorption frequency. We have added text to the Supplemental Note 1, reading: *"In all cases, the analytical model predicts a response close to ν_{01} , but the experimental intensity of transmitted light is below the detection limit of our instrument and we cannot measure a transient signal."*

Supplemental, line 94: close to 1
Changed as recommended.

Supplemental Note 2: add references on the computational method for DFT calculations - line 111, PCM is not defined for example; equation S9 uses 5 parameters that seem to be determined from less than 5 data, what did the authors do?

We modified line 111 from Supp. Note 2 to include the following with the reference (13) added: *"The reduced mass of the NO stretching mode is 14.75 AMU, obtained from a normal mode calculation on an optimized geometry of SNP at the CAM-B3LYP/6-311++G** level of theory.[SI Ref 13] The polarizable continuum model (PCM) tuned with the dielectric constant of methanol ($\epsilon = 32.613$) is included to account for the solvent electrostatic interactions"*.

Supplemental Table 5: the experimental UP2->MP2 can be added.

Changed as recommended.

Reviewer 3

1) The authors should show a fit to the Rabi oscillations in Figure 3b, indicating the frequency and dephasing times extracted from their fit.

Rev.3; Point 1. Per the reviewer request, we fit the Rabi oscillations in Fig. 3b and have updated the figure to include the fit. In the manuscript we added the following text to pg 5: *“When the full pump is employed, the transient response reveals oscillations which have a period of 650 fs and a dephasing time of 700 fs based on the results of fitting to a damped sinusoidal wave, as shown in Fig. 3b.”*

2) In some of the figures it is difficult to discern the frequencies that are being discussed. For key frequencies such as 1931 cm⁻¹ it would be helpful to indicate, with a light dashed line or arrow, the features being discussed. In Supplemental Figure 3 vertical lines at the peak positions would help the reader see that there are no observable frequency shifts.

Rev.3; Point 2. We have made the suggested change to Supp. Fig. 3 and thank the reviewer for the suggestion. Furthermore, to assist with identifying specific modes of interest, especially in the subtracted pump-probe spectra, we have increased the tick density on the plots and labeled each feature of interest i-v to facilitate discussion in the text.

3) A key claim in this paper is there exists an incoherent population in the UP level. This claim is supported by the data shown in Figure 5, where the UP to UP2 kinetics are fit to a biexponential with rates of 1ps and 8ps. The authors argue because the 8ps relaxation is considerably longer than the 2ps they report for the polariton dephasing time, there must be an incoherent population in the UP level. In Supplemental Figure 8 they show similar rates for the LP. Why is the emphasis placed on the UP in the main text? Figure 5 should give a more extensive comparison of the various population kinetics.

Rev.3; Point 3. As we describe in the response to Reviewer 2's Point 4, the features in the LP-Probe region overlap with features associated with reservoir population making their analysis somewhat ambiguous. The LP features have qualitatively similar decays but we have not arrived at a definitive assignment of their origin. We have added text to pg 10 that reads: *“The other features, both positive (iii and v) and negative (ii and iv), decay with indistinguishable kinetics (with decay times 6.5 to 8 ps, see Supp. Fig. 8 and Table 6), leading us to speculate that the decay time reflects an inherent property of the polaritons in this system. The proximity of the features in the LP Probe region to the reservoir excited-state absorption introduces considerable spectral ambiguity and precludes a definite assignment of those features.”*

4) The authors state “The specific mechanism of polariton population decay remains a topic for future study”, and “Our new observations may have implications for designing photophysical experiments that rely on excited state population in vibration-polariton modes.” While future experiments are surely needed, the authors should share some of their thoughts on these important issues. They should comment on the possible mechanisms of polariton population decay and on the implications of their findings on future applications employing polaritons for controlling excited state populations and modifying chemical reactivity.

Rev.3; Point 4. Indeed, the specific mechanisms of polariton decay are important and will likely influence important cavity-induced phenomena (as pointed out by reviewer 1, comment 2). Although

the results reported in this work do not provide sufficient information for identifying the polariton relaxation paths, we are able to make the following discussion, added to p. 11 of the revised manuscript. *“While the results reported here are important in the sense that distinct coherent and incoherent polariton lifetimes are identified, the specific mechanisms of polariton decay are not clear. Identifying the relaxation processes are, of course, key to developing approaches to controlling these lifetimes. We hypothesize that polariton coherence and population times are important for enabling energy transfer processes, such as those reported in Ref 27, and influence the decay of transition state species, which could be a mechanism at work in modified chemical reactivity. Polariton decay is certainly a prominent concern when forming polariton condensates, which has yet to be achieved with vibrational polaritons. Additional studies of polariton relaxation, which aim to contribute to these areas, are currently under way.”*

Reviewer #1 (Remarks to the Author):

The authors have responded constructively to all the referees comments and I think the paper is now ready for publication.

Reviewer #2 (Remarks to the Author):

The manuscript reports very relevant and interesting results on the dynamics of vibrational polaritons. The authors have greatly improved the paper in its revised version. Additional work confirms the results, and their analysis is now quite convincing. They have carefully taken into account the suggestions of the referees.

I recommend the publication in Nature Communications. I have noticed only minor points that need to be clarified before publication (see below), without the need for further revision.

Minor points:

1) There are some discrepancies between frequencies values along the text:

- excited state absorption at 1884 cm⁻¹ (p3), at 1888 cm⁻¹ (p9 and Fig.5);
- the solid colored lines in Fig. 2 are not drawn at the frequencies mentioned in the text: about 1912 cm⁻¹ for the reservoir absorption (1909 cm⁻¹ in the text), about 1931 cm⁻¹ for UP (1928.5 in Table S4, in contradiction with the UP1->UP2 transition at 1929 cm⁻¹ in the same table when the latter frequency should be lower than the GS-UP1 transition), about 1896 cm⁻¹ for LP, which gives a Rabi splitting of 35cm⁻¹, in contradiction with 40cm⁻¹ mentioned in the caption
- 1928 cm⁻¹ and 1878 (or 1880) cm⁻¹ are shown as UP and LP frequencies (respectively) in Fig. 3 and Fig. S4, i.e. different values from those in Fig. 2 and Fig.4 for the same transitions. Does this correspond to a different experiment with different cavity characteristics?

2) Figure 3b: "Intensity at the probe frequency" should be modified in "Delta A", as in Figure S4, in agreement with other figures.

3) Page 11: "This is clear when examining Figs.5, S1, and S2"

Reviewer #3 (Remarks to the Author):

The authors have thoughtfully adequately addressed the concerns of all of the reviewers. I now recommend publication of this manuscript in Nature Communications.

Reviewer #1 (Remarks to the Author):

The authors have responded constructively to all the referees comments and I think the paper is now ready for publication.

No changes required.

Reviewer #2 (Remarks to the Author):

The manuscript reports very relevant and interesting results on the dynamics of vibrational polaritons. The authors have greatly improved the paper in its revised version. Additional work confirms the results, and their analysis is now quite convincing. They have carefully taken into account the suggestions of the referees.

I recommend the publication in Nature Communications. I have noticed only minor points that need to be clarified before publication (see below), without the need for further revision.

Minor points:

1) There are some discrepancies between frequencies values along the text:

- excited state absorption at 1884 cm⁻¹ (p3), at 1888 cm⁻¹ (p9 and Fig.5);

- The value 1884 cm⁻¹ on p. 3 is for the excited state absorption outside the cavity, which is correct, while the one on p. 9 and Fig. 5, at 1888 cm⁻¹, is for the probing the reservoir excited state absorption through the LP transmission, which is near but slightly modified compared to outside the cavity. This small shift is characteristic of the strong-coupled response and captured by the analytical calculations for the transient transmission response.

- the solid colored lines in Fig. 2 are not drawn at the frequencies mentioned in the text: about 1912 cm⁻¹ for the reservoir absorption (1909 cm⁻¹ in the text), about 1931 cm⁻¹ for UP (1928.5 in Table S4, in contradiction with the UP1->UP2 transition at 1929 cm⁻¹ in the same table when the latter frequency should be lower than the GS-UP1 transition), about 1896 cm⁻¹ for LP, which gives a Rabi splitting of 35cm⁻¹, in contradiction with 40cm⁻¹ mentioned in the caption

	Theory [cm ⁻¹]	In Experiment [cm ⁻¹]	$ \langle j \hat{a} i\rangle ^2$
GS → LP ₁	1894.0	1888.5 1896.0	0.608
GS → UP ₁	1932.0	1928.5 1931.0	0.392
LP ₁ → LP ₂	1888.0	1896.0	0.825
LP ₁ → MP ₂	1920.0	-----	0.777
LP ₁ → UP ₂	1964.0	-----	0.006
UP ₁ → LP ₂	1849.0	-----	0.008
UP ₁ → MP ₂	1881.0	1886.0	0.306
UP ₁ → UP ₂	1926.0	1929.0	1.077

We modified fig.2 so the line for the reservoir is at 1909 cm⁻¹ to match the text. Additionally, we changed the experimental value in Table S4 for GS-> UP₁ to 1931 cm⁻¹ and GS-> LP₁ to 1896 cm⁻¹. We also changed the last line in caption for Fig. 2 to read: "The Rabi splitting in this data is 35 cm⁻¹." and the caption for Supp. Table 4 to read: "Experimental value = 35 cm⁻¹."

- 1928 cm⁻¹ and 1878 (or 1880) cm⁻¹ are shown as UP and LP frequencies (respectively) in Fig. 3 and Fig. S4, i.e. different values from those in Fig. 2 and Fig.4 for the same transitions. Does this correspond to a different experiment with different cavity characteristics?

Yes, the sample for Fig. 3 is a slightly different concentration with a different Rabi splitting than those in Fig. 2 and Fig. 4. Fig. 3 sample/cavity is closer to the one used for Figure 4. We mention in the text (a few lines above Fig. 3 on p. 5) that the splitting in the case for Fig 3 is 45 cm⁻¹ which is closer to the one for Fig. 4 (46.5 cm⁻¹; this value is stated in the Supp. Table 5 and was incorrect in the Fig. 4 caption but is now fixed to indicate the same value: 46.5 cm⁻¹). Also, we have a statement in the Methods section that justifies the different Rabi splitting among our samples, "Filling the cavity with the saturated SNP solutions gives a Rabi splitting, $\Omega \sim 35-45$ cm⁻¹. The Rabi splittings varied due to using slightly different concentrations among our samples." We changed the range for "40-45" to "35-45" to include the lower splitting value for the sample in Fig. 2.

2) Figure 3b: "Intensity at the probe frequency" should be modified in "Delta A", as in Figure S4, in agreement with other figures.

The figure was modified so the y-axis label is now ΔA like the other figures.

3) Page 11: "This is clear when examining Figs.5, S1, and S2"

Changed as recommended.

Reviewer #3 (Remarks to the Author):

The authors have thoughtfully adequately addressed the concerns of all of the reviewers. I now recommend publication of this manuscript in Nature Communications.

No changes required.